# Performance barriers of Civil Registration System in Bihar: An exploratory study

Krishna Kumar[1], Nandita Saikia[2]*, Nadia Diamond-smith[3]

1 Centre for the Study of Regional Development, School of Social Sciences III, Jawaharlal Nehru University, New Delhi, India, 2 Department of Public Health and Mortality Studies, International Institute for Population Sciences, Deonar, Mumbai, India, 3 Epidemiology and Biostatistics Department, University of California, San Francisco, CA, United States of America

* nanditasts@gmail.com

## Abstract

### Objectives

Vital statistics generated by the Civil Registration System (CRS) are essential for developing healthcare interventions at all administrative levels. Bihar had one of the lowest levels of mortality registration among India's states. This study investigates CRS's performance barriers from the perspective of CRS staff and community members in Bihar.

### Methods

We conducted a primary qualitative survey in the two districts of Bihar during February-March 2020 with CRS staff (n = 15) and community members (n = 90). We purposively selected the Patna and Vaishali districts of Bihar for the survey. Thematic analysis was done to identify the pattern across the data using the Atlas-ti software.

### Results

Most participants showed a good understanding of registration procedures and birth and death registration benefits. The perceived need for death registration is lower than birth registration. Birth registration was higher among female children than male children. We found that most participants did not report children or adult female death due to lack of financial or property-related benefits. Most participants faced challenges in reporting birth and death due to poor delivery of services at the registration centres, higher indirect opportunity cost, and demand of bribes by the CRS staff for providing certificates. We found a lack of adequate investment, shortage of dedicated staff, and limited computer and internet services at the registration centres.

### Conclusions

Poor data on birth and death registration could lead decision-makers to target health services inappropriately. Strengthening health institutions' linkage with the registration centres, mobile registration in far-flung areas and regular CRS staff training could increase death

**Data Availability Statement:** All relevant data are within the manuscript and its Supporting Information files.

**Funding:** The author(s) received no specific funding for this work.

**Competing interests:** The authors have declared that no competing interests exist.

registration levels. An adequate awareness campaign on the benefits of birth and death registration is required to increase the reporting of vital events.

## Introduction

A Civil Registration System (CRS) is a permanent, continuous, compulsory, and universal recording of vital events with the legal requirements [1]. The Civil Registration Vital Statistics is exact and actual data, certified by registering authority, and legally acceptable [2]. A birth certificate documents information such as age, place of birth, and family background [3]. Besides official documentation of a child's birth, it facilitates government-aided essential services such as education, health facilities, and social security [4–6]. CRS also provides timely and reliable vital statistics. Health officials and policymakers mainly use CRS data to track fertility, mortality, and epidemiological patterns at all administrative levels. United Nations documented that a functional CRS is the best source of information on vital events for administrative, demographic, and epidemiological purposes [7].

Despite government agencies and UNICEF's effort to universalize birth registration globally, about 166 million children under age five and 40 million infants were not officially documented [3]. According to UNICEF's report (2019), about 77 million children under age five do not have a birth certificate in South Asia. There has been a significant disparity among countries in terms of birth registration. Low or middle-income countries showed more unsatisfactory performance in registering the child's birth. Mortality statistics are also essential for identifying health risks and evaluating health programs. In many countries, the completeness of death registration is lower than birth registration. World Health Organisation (WHO) recorded about the two-thirds of global deaths were not registered. Besides, most WHO members obtained incomplete data on mortality and the cause of death [8]. Another study showed that administrative challenges, insufficient technical capacities, and unawareness are significant challenges in death registration [9]. Accurate recording of death's cause is challenging in lower and middle-income countries [10, 11].

In India, the Births, Deaths, and Marriage Registration (RBD) Act, enacted in 1886, suggested voluntary registration of births and deaths. However, it was not uniformly implemented across India. After independence, India's Government introduced the Registration of Births and Deaths Act in 1969, which mandates registration of all births and deaths within 21 days [2]. A low birth registration is associated with increased school dropout, child trafficking, and child labor [1, 12]. A higher child mortality rate is associated with higher level of unregistered children [13]. A lower death registration limits medical research and reliable mortality estimates [2, 14]. Despite India's mandatory birth and death registration law, according to official statistics, nearly 7 percent of births and 8 percent of deaths were not registered in India in 2019 [2]. Further, there are disparities at the state and district level regarding coverage and completeness of birth and death registration. In India, seventy percent of the districts have more than a million population each; lower death registration coverage at the district level precludes estimating health indicators with precision [15]. Universal civil registration in the interest of sound policymaking and its implementation helps monitor the SDGs goals such as promoting healthy lives and well-being, reducing mortality rate and legal identity for all [16, 17].

Bihar is one of the least developed and third most populated state of India. A comparison table of demographic and health indicators of Bihar and India is shown in S1 Table. The total population of Bihar is 113 million (which is equivalent to the population size of Ethiopia, the

world's 12[th] populous country), constituting 8.57% of India's population. According to the National Family Health Survey, 2015–16, Bihar has the highest total fertility rate (3.14) in India. The infant death rate and under-five mortality in Bihar are 48.1 and 58.1 per thousand live births. Education attainment is quite low, with only 14 percent of the household population of eighteen and above completed 12 or more years of schooling. Among the children, age 0–6 years, 41 percent and 33 percent received immunisation and health check-up from an anganwadi sevika. The percentage of institutional birth is lower in Bihar (64%) as compared with India level (79%) [18]. In Bihar, birth and death registration levels are much lower than the national average. However, there has been an increase in birth registration levels in Bihar from 64% in 2014 to 89% in 2019 [2, 19]. On the other hand, the death registration level was lower (52%) in Bihar compared with other Indian states in 2019 [2]. Additionally, Bihar showed inconsistency in death registration level from 43% in 2017 to 35% in 2018 and further, the level increased to 52 percent in 2019 [2, 20, 21].

There has been increased research around the impact of under-registration, but existing studies focused on the need and benefits associated with functional registration systems [6, 22, 23]. Recent studies primarily focused on birth and death registration levels, trends and estimation of death registration level using indirect methods [24–26]. However, such studies did not answer why death registration is lower in some states of India. While media reports and public health experts highlighted the importance of up-to-date vital statistics for initiating healthcare intervention during a health emergency, peer-reviewed literature investigating motivations and barriers to birth and death registration in India is limited. Previous studies suggested qualitative studies in the states or districts where a lower registration level was recorded to explore the contextual determinants of under registration [7, 9, 27]. This study investigates the performance barriers of the CRS in Bihar, a state with the lower level of birth and death registration. This study provided crucial inputs to reform administrative issues for an effective CRS in Bihar.

## Materials and methods

We conducted a primary qualitative survey in the two districts of Bihar during February-March 2020 with CRS staff (n = 15) and community members (n = 90). Fig 1 shows the framework of the operation of CRS in Bihar.

### Study design and sample

Fifteen CRS staff and ninety community members participated in the survey. Out of 38 districts in Bihar, we purposively chose two districts Patna and Vaishali, from which we randomly selected four blocks from each district. Fig 2 shows the study area. We chose Patna and Vaishali districts to investigate how registration practices differ in two different contexts. These districts represent two different urbanization contexts and registration levels: Patna being the most urbanized (43%) and Vaishali the least urbanised district (7%) of Bihar. Patna has better health care facilities, a better road network, CRS offices equipped with modern technologies, death registration level (61%) and a higher literate population (71%). On the other hand, Vaishali has a poor road network and transport facilities, lower access to public and private health institutions, death registration level (45%) and a lower proportion of the literate population (67%) S3 Table. We selected Patna Sadar, Sampatchak, Phulwarisarif, and Danapur blocks from Patna, and Hajipur, Vaishali, Desri, and Raghopur blocks from Vaishali. 4 out of total 22 blocks of Patna were randomly selected using a RANDBETWEEN function in excel. Similarly, 4 out of 16 blocks of Vaishali were selected S2 Table. We also showed estimated birth and death registration level by districts of Bihar S3 Table.

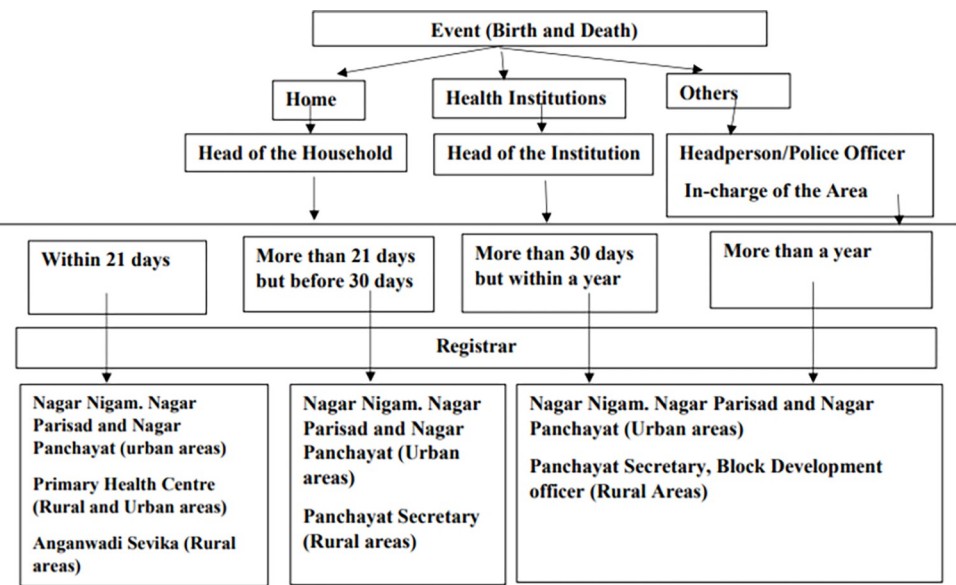

**Fig 1. Flow chart of birth and death registration in Bihar.** CRS annual report, 2020. Department of Planning and Development, Bihar, 2021. Note:- In case of birth and death registration after 30 days but within 1 year, registration is done after written permission of the prescribed authority and on the submission of an affidavit made before a notary public and payment of a late fee of rupee five (US$ 0.07). In case of registration of birth and death after an year, an order is required by a magistrate of the first class (Block Development Officer) after verifying the correctness of the event and payment a fee of rupee ten (US$ 0.14).

## Study participants

Block Statistical Officer (BSO), a Medical Officer, a Panchayat Chief or Ward Parsad (local leader), an anganwadi sevika and a registrar at Nagar Nigam from each block were chosen for the key informant interview (KII). An anganwadi sevika is a community health worker from the local village who delivers services to pregnant and lactating women and children below the age of six years through anganwadi centres (AWCs), serving a catchment area of a population of 1,000 [28, 29]. We selected the key informants based on their position, working experience, availability, and consent for the interview. Further, we selected 8–12 people from each block for Focus Group Discussion (FGD). The inclusion criteria for FGD included: any community member aged between 18–65 years who belong to the household where birth or death took place in the last five years and consented to participate in the study.

## Data collection

KK and NS developed semi-structured questionnaires and selected the study area. KK conducted key informant interviews (n = 15) and focus group discussions (8 FGD with 90 participants) with the local person's assistance to access the study area. KK went to block offices, primary and community health centres, Panchayat chief or Ward Parsad (local leader) office, and anganwadi centres, accessible to him. In addition, Assistant Registrar or BSO, Medical Officers and registrars at Nagar Nigam were invited for the interview. Fig 3 shows the hierarchy of CRS staff in Bihar. We also did telephonic interview with two CRS staff (Block Statistical Officers) to assess the operational status of the system during the pandemic.

Further, KK visited households with assistance from a Panchayat (a village council) Secretary or Ward Parsad and volunteer (postal worker, anganwadi sevika and teacher) of the area and invited an eligible household member (male or female) to participate in FGD. An eligible

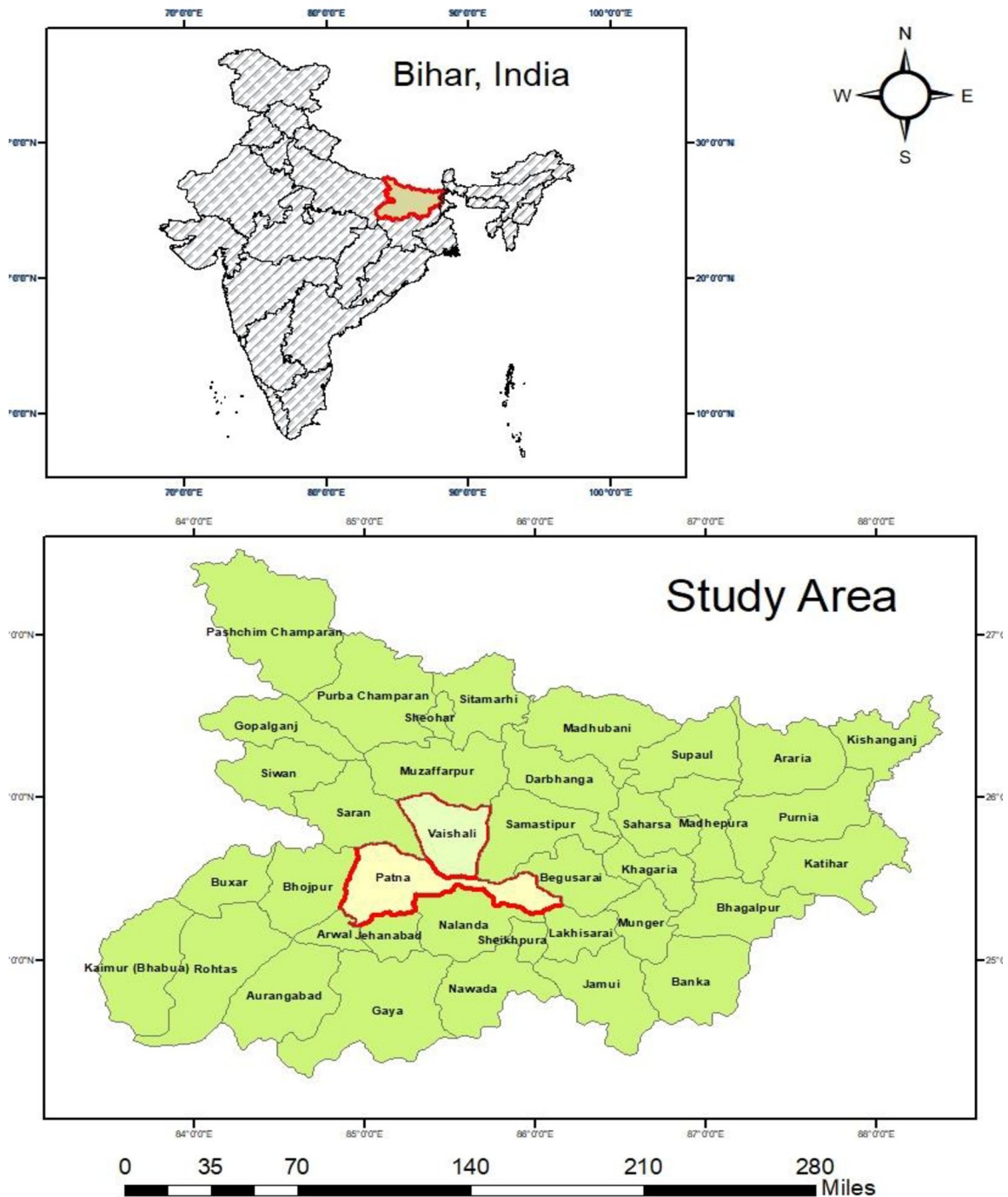

**Fig 2. Study area.** Source- Authors generated the map using GIS.

household member is a male or female adult of the household (18 years age and above) who can respond on behalf of the household. KK conducted 8 FGD, 1 FGD in each of the selected blocks, including 10–12 community members. KK noted all information during interviews with key informants to ensure no essential points were missed. In addition, KK took consent for note-taking and audio recording from all participants. A few community members (20

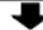

> **State level**
> - Chief Registrar: Director of Directorate of Statistics and Evaluation, Planning and Development Department
> - Joint Chief Director: Joint Director (Administration, Planning and Development Department)
> - Dy. Chief Registrar: Dy. Director (Vital Statistics, Planning and Development Department)

> **District and block level**
> - District Registrar: District Magistrate (Birth and Death)
> - Additional District Registrar: District Statistical Officer (urban areas) and Block Development Officer (rural areas) (Birth and Death)
> - Registrar : Chief Medical Officer/Executive Officer/Special Officer (Birth and Death, Nagar Nikay)
> - Registrar: Deputy Superintendent Officer of Medical College/Sadar Hospital/ Sub-Division Hospital (Health Department)

> **Local Area Level**
> - Registrar: Panchayat Secretary (rural areas) (Ministry of Panchayati Raj); Medical Officer in charge of Primary Health Centre or Referal Hospital (urban areas and rural areas) (Birth and Death) (Health Department)
> - Sub-Registrar: Anganwadi Sevika (rural areas) (Integrated Child Development Service, Social Welfare Department)

**Fig 3. Hierarchy of Civil Registration System in Bihar.** Source- CRS annual report, 2020. Department of Planning and Development, Bihar, 2021.

people) refused to participate, citing their reasons, and therefore, KK excluded them from participation. Each FGD consisted of a mix of male and female respondents. The demographic profile of all the study participants is shown in Table 1.

**Table 1. Demographic profile of the study participants.**

| | Mean age (in years) & (range) | Study Area | | | | Number of respondents | Method of data collection |
|---|---|---|---|---|---|---|---|
| | | Patna | | Vaishali | | | |
| | | Male | Female | Male | Female | | |
| **Assistant Director** | 50 | 1 | - | - | - | 1 | KII |
| **System Manager** | 35 | 1 | - | - | - | 1 | KII |
| **BSO** | 41 (40–42) | 1 | - | 1 | - | 2 | KII |
| **Anganwadi Sevika** | 44.3 (40–50) | - | 2 | - | 2 | 4 | KII |
| **Medical officer** | 52.2 (45–62) | - | 2 | 1 | - | 3 | KII |
| **Ward Parsad** | 29.5 (24–35) | 1 | - | 1 | - | 2 | KII |
| **Nagar Nigam** | 36 (30–42) | 1 | - | 1 | - | 2 | KII |
| **Community members** | 35.6 (17–62) | 30 | 13 | 25 | 22 | 90 | FGD |

Question schedules and topic guides were designed using previous literature [8, 23, 30, 31]. We used an open-ended questionnaire to capture informants'experiences, stories, ideas, and case studies S1 Text. Questions were asked in the Hindi language. KK collected the data. KK also received seven days of training on the qualitative data collection method before the primary surveys. The confidentiality of participants fully adhered.

### Reflexivity

At the time of conducting this study, the interviewer KK was an M.Phil. Student. The interviewer belonged to the Patna district of Bihar. KK first stressed his identity as belonging to the same state and speaking the same language (Hindi). KK built familiarity and rapport with the study participants. Local persons helped him in finding the eligible participants for the study. KK explained the study's objective to them, and gave assurance that the researcher had no personal ulterior motives for conducting the survey. KK did not pre-assume any results and entirely relied on the participants' spoken words.

### Data analysis

We anonymised all recorded notes and audio. We chose thematic analysis for obtaining a systematic framework to code qualitative data to identify patterns across the data [32]. The transcripts were transcribed in the local language (Hindi). Further, the recorded transcripts were translated into English to understand the theme by the wider population. KK coded transcripts and organised them by the sub-group of the participants. We used qualitative data software Atlas-ti 8.0 for the thematic analysis of data. We used direct quotes for exemplary purposes. The senior author (NS) reviewed typed transcripts for accuracy, completion, and plausibility. We also looked for the data saturation to validate our findings.

### Ethics statement

We obtained informed consent from all the participants before any interviews and group discussions were conducted. The study protocols have been reviewed and approved by the institutional ethical review board at the Jawaharlal Nehru University (Ref. No. 2019/Student/235).

## Results

### Themes generated based on FGDs among community members

Based on the FGDs, we identified three main themes that were found to be critical barriers in well functioning of the system at the community level: 1) Low awareness and inadequate knowledge of registration procedures and the birth and death registration benefits, 2) people's attitude and perception towards birth and death registration and 3) administrative challenges experienced by the community members.

**1. Low awareness and inadequate knowledge of registration procedures and the birth and death registration benefits.** Majority of FGDs' participants knew how to apply for birth and death registration. Among total FGDs' participants (n = 90), nearly fifty percent registered their child's birth to anganwadi sevika and block officers. We found that a majority of study participants or their household female members (60 out of 90) delivered a baby in a public hospital. However, a subset of these participants (20 out of 60) reported their child's birth to anganwadi sevika, block officials and Nagar Nigam offices due to no availability of dedicated staff and shortage of registration forms at the public health facilities. In addition, requirement of identity cards was found to be perceived barriers to birth registration. According to the participants, identity proof (Aadhaar card, residential certificate, and ration card) is required for

identity verification before birth and death registration. In the case of a child's birth registration, his or her parent's identity proof is required.

Aadhaar is a 12 digit individual identification number issued by the Unique Identification Authority of India on behalf of India's Government. This card serves as proof of identity and addresses anywhere in India. The district court issues a residential certificate about the place of birth and residence of the individuals. A ration card is also an official document issued by state governments in India to households eligible to purchase subsidised food grain from the Public Distribution System under the National Food Security Act. This card also serves as a common form of identification for Indian citizens.

> *"I received a birth certificate from the block office. My daughter's birth certificate was required for admitting her to school. Anganwadi sevika also demands my daughter's birth certificate for providing mid-day meal and benefits of government schemes such as financial assistance to the girl child."* (25 years old female, Patna)

> *"I applied for my son's birth registration three months ago. My son got admission to the school without a birth certificate, so I did not go to the block for collecting his birth certificate. I also know a birth certificate is required to vaccinate him at a public health facility. I will go to the block office to collect the birth certificate when I get time."* (32 years old man, Vaishali).

The registration law mandates all birth and death registration within 21 days. However, two-fifth of the total participants (n = 90) applied for their child's birth registration after one year. A few participants mentioned that they applied for a child's birth registration when there was a requirement of a birth certificate for their child's enrolment in school. According to respondents, delayed registration is allowed on payment of five rupees (US $0.07) as delayed registration fees and an affidavit made before a notary public. A birth certificate is required for receiving benefits of government-aided services such as vaccination, financial support after a child's birth, and years of formal education. Birth registration was higher among female children because the Bihar government has initiated financial assistance scheme for girl children. We found that less than half of the study participants received birth certificates within one month. However, a subset of participants (10 out of 90) did not receive a birth certificate for their child at all.

> *"A medical staff of a private hospital informed me that 2000 rupees would be deposited on my daughter's name by the government after her birth registration, after that, I reported my daughter's birth to a registrar at Nagar Nigam and requested a birth certificate."* (25 years old female, Vaishali)

> *"I applied for my daughter's birth registration after two years of her birth. I wanted to register her birth after her name ceremony. I paid ten rupees as delayed registration fees and requested the order of the District Magistrate for her birth registration." (*35 years old man, Patna*)*

Unlike birth registration, fewer participants (50 out of 90) report the death of their relatives or household members to civil authorities. A few participants (15 out of 90) reported death at the registration centres situated at cemeteries or burial places. Besides, two-fifth of the study participants (36 out of 90) reported the death of their household members at the block or municipality offices. However, rural respondents said that there is no registration centre at the cemetery and burial place in their village; therefore, they are required to report to anganwadi sevika, panchayat secretary or block officials. A subset of participants (50 out of 90) said they submitted the deceased's Aadhaar card as identification proof. Most participants (75 out of 90)

knew that a death certificate is mainly required to seek family pension, claim for insurance, and transfer property after a property holder's death.

*"I reported the death of my uncle to the block office. A death certificate is required to seek a family pension, claim for insurance, and transfer property to another name after a property holder's death. Reporting death is also useful for medical officers to study the cause of death."* (26 years old woman, Patna)

Nearly one-sixth of the total participants (15 out of 90) were unaware of the death registration procedures and benefits associated with death registration. A higher number of participants (35 out of 50) from Patna applied for birth and death registration of their household members than participants (19 out of 40) from Vaishali district. Rural and urban differentials in birth and death registration were found. Sometimes, people did not report births due to the lack of necessary documents or lack of support from the CRS officials.

*"I did not know about the documents required for a child's birth registration. Also, I did not find anyone who could assist me in submitting the registration form at the block office; therefore, I did not report the birth of my daughter yet."* (28 years old woman, Vaishali)

**2. People's attitude and perceptions on birth and death registration.** There was a high demand for birth certificates among community members. The majority of participants (70 out 90) reported birth registration of their household younger members because many immediate benefits are associated with birth registration. Nearly two-fifth of participants (36 out of 90) reported for their child's birth registration after a year due to traditional norms or local customs such as naming ceremonies. We found that participants gave more importance to the naming ceremony than the formal registration of the child's birth.

Most participants (50 out of 90) said that there was lower importance of death registration than birth registration. The death certificate was not required before cremation or burial. Some participants (40 out of 90) perceived death registration should be done within the prescribed time. A few participants (15 out of 90) also said it is not essential to report infant death whose birth was not registered. A few participants (10 out of 90)) did not report infant deaths because they believed that infant death is associated with sin of previous birth, and it should not be disclosed to others.

*"In our tradition, a father cannot fire a dead body of his children. Death of an infant is considered a sin of parents' previous birth; therefore, people hide infant death from society members and hence, don't report".* (38 years old man, Vaishali)

One important reason for not registering children's death was that there was no perceived necessity when the deceased did not possess property in their name. Also, there are no monetary or administrative incentives for registering deaths.

*"I was aware that death is reported at the block or municipality office. However, I did not report my child's death because there were no benefits attached to it. Also, there was no property registered on his name."* (35 years old man, Vaishali)

*"I was aware that death registration is done at the block office and Nagar Nigam. However, I did not report the death of my wife to a civil registrar because a death certificate was not required as there was no property registered on her name."* (40 years old man, Patna)

**3. Administrative challenges experienced by community members.** About less than half of total respondents (40 out of 90) of FGD revealed that they faced problems in reporting their child's birth and household member's death. Some participants stood in a long queue for 3–4 hours at the blocks and Nagar Nigam offices. A few participants reported no water facility and shading for the people at the registration offices. Most of the participants (35 out of 90) took half day leave from their work to report their child's birth. Nearly 20 participants were asked to come the next day due to irregular electric current and no printing materials at the block offices. Besides, we found nearly two-fifths of the participants (35 out of 90) paid 50–200 rupees bribe to anganwadi sevika and block officers for receiving the birth or death certificate. However, a birth or death certificate should be provided free of cost by law. A few participants (10 out of 90) travelled 2–4 hours to reach registration offices and lost their wages for a day.

*"An officer told me there is no fixed time to provide the certificate. He also asked for 100 rupees for providing my son's birth certificate."* (23 years old man, Patna)

*"An anganwadi sevika registered my daughter's birth, but she demanded 100 rupees for providing the birth certificate. I did not have money at that time, so I did not collect the certificate. Later, I ignored her because I was not willing to give her 100 rupees for the birth certificate." (*30 years old man, Patna*)*

*"I travelled 2 hours to reach the block office and stood 2–3 hours at the block office to report my father's death. I lost wage for a day and spent 100 rupees on travelling."* (38 years old man, Vaishali)

## We generated three themes based on KII with CRS staff: 1) Awareness and attitudes of the CRS staff, 2) Administrative and technical challenges, and 3) Staff's suggestions to improve birth and death registration level

**1. Awareness and attitudes of the CRS staff.** Among 15 staff interviewed, 12 staff showed good knowledge of laws and protocols associated with birth and death registration. They revealed birth and death certificates are provided free of cost, given registration was done within 21 days of the event. Moreover, delayed registration is allowed on submission of the affidavit, late fine and block and district level officers' approval. We found most of the staff perceived timely and complete birth and death registration is crucial for generating accurate vital statistics for policymaking. However, half of the total CRS staff interviewed did not read the CRS manual.

*"I enter birth and death data into the system. A few times I feel difficult to assign correct codes in the birth and death registration forms due to unavailability of latest CRS manual in the office.*" (a CRS staff, Patna)

**2. Administrative and technical challenges.** Despite the government's announcement of the CRS portal for birth and death registration at a real-time basis in all the registration units, we found 1 out of 4 block offices of Patna, and 3 out of 4 block offices of Vaishali were not equipped with modern technologies such as computers, printers, photocopier and specific CRS software. Irregular electric supply and no separate department for CRS were observed. We also found many staff of other department performed the duties of birth and death registration besides their departmental work.

*"I am often given duty as an invigilator during the state's secondary board examination and other examination for screening of the candidates for public sector's jobs. I am also directed to*

*assist officers in the conduct of municipalities or the state-level election."* (a block-level officer, Patna)

More than half of the total staff interviewed (10 out of 15) mentioned that targets were provided for birth and death registration by the office of Registrar General of India. The target for registration coverage is higher for births (>90%) than death registration (80%). These targets are the estimated number of birth and death to be registered by the CRS staff in a particular month. CRS manual shows the mathematical formula for calculating such a target. They also expressed that most of the medical attendant was not skilled in recording the cause of death.

*"I received a monthly target of birth and death registration by the state headquarter. I don't know the mathematical formula for calculating the target. Also, I don't believe in such a target as institutional birth data are transferred directly to the Nagar Nigam and then district headquarter and not shared with me. Many times, the target is unreliable and difficult to achieve it".* (a CRS staff, block office, Vaishali).

This study found an anganwadi sevika performs multiple tasks such as providing pre-natal and post-natal care, assistance during birth delivery, and birth and death registration. They also teach children enrolled at their centre and serve mid-day meals to them. The pressure of performing multiple tasks limited their role as a sub-registrar and affected the quality of registration.

*"Block officers assign multiple tasks to me. I am required to run an anganwadi centre where I teach and serve mid-day meals to the children. I also register birth or death occurs in this area."* (an anganwadi sevika, Vaishali)

Furthermore, there was no regular training provided to the CRS staff. We found most of the CRS staff did not receive training in last one year. Anganwadi sevika was not invited in the block level meeting since last year. Besides, they were not provided with a birth and death registration forms and honorarium on time. In addition, there was also no adequate fund available for the periodic training and workshops for the staff.

*"Block-level officers don't call me in the review meeting. I think monthly, or quarterly review meeting is required for solving problems encountered during birth and death registration."* (an anganwadi sevika, Patna)

*"I have not received any training in the last two years. I attended a training program on organising workshops on awareness on birth and death registration benefits two years back. I need training on how to operate CRS portal on a computer system."* (a block-level officer, Patna)

There was no regular supervision at the block level or community level by the district or state level officers. Additionally, there was no strict implementation of the RBD Act, and the officers did not take any steps for improving the registration level in recent times, particularly death registration. We found health institution does delayed registration; however, it is not allowed as per official guidelines. We found poorer functioning of the registration system in the block offices of Vaishali as compared to the block offices of Patna.

**3. Staff's suggestions to improve birth and death registration level.** Most staff (10 out of 15) interviewed revealed that regular awareness programs on birth and death registration benefits could increase registration levels. Posters and banners exhibition during most celebrated festivals like Durgapuja, Chhat Puja, Ramnavmi procession, Harihar Kshetra Mela (Sonepur

cattle fair which attracts visitors from all over Asia) was widely suggested by the officials and staff. A few staff (5 out of 15) told mobile registration in hard-to-reach areas, or the villages far from registration centres could improve birth and death registration levels.

Further, medical personnel associated with CRS revealed that the birth and death reporting had increased after CRS' association with health facilities. Besides, the system manager, an assistant director, and block-level officers revealed that adequate awareness campaign on importance of death registration is needed to increase death reporting. In addition, they suggested that adequate funds should be provided for the smooth functioning of the system.

We also interviewed two CRS staff over telephone after third wave of Covid (January 2022) in India. They revealed that they did not achieve the birth and death registration level target due to shortage of staff. Majority of staff were appointed in Covid management duty. Therefore, the system failed to record birth and death on time during the pandemic.

## Discussion

The present study investigated CRS's performance barriers in a poor state of India where birth and death registration is low. In 2012, there had been lower recording of birth and death. Again, there was slowdown in registration due to revamping of system in 2016 from paper-based registration to online system (CRS portal) for registration of vital events. Moreover, in order to accelerate the vital statistics quality and coverage and to create the awareness among the public, a descriptive analysis was published in the quarterly report of Directorate of Economics & Statistics namely "Bihar Sankhyaki Darpan". The message on benefits of birth and death registration were also telecasted from Doordarshan Patna to attract the attention of rural and urban people towards registration [33]. In 2019, the completeness of birth and death registration level in Bihar was 89.3 and 51.6 percent respectively [2].

The inadequate knowledge and attitude of community members on birth and death registration procedures and benefits are barriers to registration levels. Previous studies documented unawareness among people is a significant challenge in birth and death registration [2, 23, 31]. In our study area, knowledge on registration was not a barrier, yet we found a discouraging attitude, particularly to death registration, contributed to incomplete CRS in the study area. Perception of little benefits of death reporting led to lower death registration in Bihar. The inadequate knowledge and perceptions of low benefits of birth and death registration limited the coverage of CRS in lower and middle-income countries [30, 34].

Most people reported their child's birth to civil authorities because there are financial and social benefits associated with birth certificates. The government of Bihar encourages birth registration and provides financial assistance to female children after their birth registration [35]. Moreover, birth certificate is required for child's school enrolment, vaccination and his social security [4, 5]. Unlike reporting a child's birth, reasons for lower reporting for a person's death are not widely known. Consistent with previous studies, our study also documented a lower registration of infant and female deaths [24, 25, 27]. According to our participants, a lower death registration among females and children is due to a lack of legal necessity and no immediate financial or property-related benefits. We also observed that the presence of social stigma for premature death is negatively impacting death registration coverage. A lower reporting of infant deaths resulted from traditional beliefs such as an infant death as the sin of parent's previous birth. For burial permission, reporting of death to civil authority was not required. Therefore, there is a lower priority for death registration among the participants. Underreporting of infant deaths that happen outside of facilities may lead to underestimating infant mortality.

In addition, differential reporting by sex will lead to insufficient quality data on sex differences in mortality rates [36]. Higher death registration of men could be associated with their engagement with the formal employment sector. Generally, family members are required to submit a death certificate to an employer to obtain social safety benefits after the death of a working member of the household. However, in our study areas, most of the respondents were engaged in the informal sector such as farming, construction of houses, a small business etc. that seldom provide social benefits to their families. Financial incentives for performing funeral activities and minimal financial assistance to deceased families may improve death reporting [37].

Even birth and death that occurred in health facilities was not recorded due to lack of dedicated staff and staff's negligence. Public facilities record all birth and death but don't update on the CRS portal until birth and death certificates are requested by family members. A study documented a similar argument for lower reporting of institutional birth and death in Bihar [38]. No strict implantation of the law is the primary reason for lower coverage of birth and death in Bihar. The previous study also revealed strict implementation and periodic audit of death registration is required [9, 31, 38, 39].

This study highlighted poor services at the CRS offices such as a longer waiting time at the registration centres and not providing the certificates on time. In addition, the demand of bribes by CRS staff is discouraging for the participants not to report birth or death. Indirect opportunity costs such as travelling costs to the registration centre, loss of wage for a day were barriers to birth and death registration. Mobile registration in far-flung areas, reducing documentation work of applicants and assisting people in generating identity cards from concerned agencies would increase birth and death registration coverage. Moreover, awareness on specific cases like the adoption of a child and the death registration of a missing person should also be provided. Anganwadi sevika, local leaders and district officials could be engaged in awareness programs with a minimal level of training to decrease negative impact of the customs or traditional beliefs on birth and death registration.

A knowledgeable and well-trained staff is essential for effective CRS [7]. This study showed staff had a good understanding of birth and death registration law and registration procedures. However, the lack of adequate training on the use of the CRS portal was a barrier to recording vital events. Department lack dedicated staff deputed solely for CRS work. Multiple tasks overburdened CRS officials led to compromise with the quality of vital statistics. The previous study also documented lack of dedicated staff, lack of training on the use of computer systems for recording birth and death was barriers to CRS functioning [9, 31]. Refresher training is also essential for updating and solving the fields' problems [7, 38].

Regular monitoring of registration unit at primary and community healthcare, may ensure accessibility and improve birth registration level. Previous studies documented urbanisation and access to healthcare facilities are associated with higher birth registration [9, 27, 31, 40]. An improvement of health facilities in Bihar is needed to improve coverage of birth and death registration. According to National Family Health Survey-2019-20, Bihar has one of the lowest maternal and child health care service utilisation within the country [41].

Telephonic interview with CRS staff reveal that during the Covid, the system failed to achieve the target of birth and death registration as there is shortage of CRS staff. Besides, staff were appointed on Covid management duty such as awareness program, monitoring and inspection of free supply of food, medicines and other healthcare items.

This study suggests some crucial points to improve birth and death registration levels. First, recurring awareness programs on birth and death registration benefits for CRS staff and community members particularly, rural residents. Customs or traditional beliefs led to lower reporting of vital event, should be changed with adequate awareness campaign. Second, there is a need

to improve accessibility to registration centres and the use of ICT to speed up the CRS work. Third, adequate funds, dedicated staff and regular monitoring and auditing of the vital statistics may improve the registration level. Fourth, standardized operating protocol should be developed for uniform processes in the system because many registration offices are not aware of CRS manual and how to use CRS portal effectively. Fifth, state-wise registration target are set by Registrar General of India, however it is based on crude birth and death rate. We recommend state, district and sub-district level target needs to be worked out using some robust scientific methodology. The aspect of migration, live birth reported by health management information system should be taken into consideration while estimating registration target.

Despite a comprehensive analysis, this study has some limitations. This study covers only two districts of Bihar; therefore, generalisation could be made for the states or districts with a similar development level with care. We did not show comparison study with CRS operations in other states with similar level of development due to limited literature. We did not collected data by socio-economic background (such as education, wealth, religion or castes) of study participants; therefore, such analysis cannot be done in this study. In addition, this study is mainly focused on barriers at the operational level and not assessed the quality of vital statistics. Moreover, we did not check the birth and death certificate owned by respondents, so there could be under-reporting or over-reporting of birth and death registration due to recall bias. Possession of a birth certificate is considered socially desirable, so there might be bias reporting of birth registration.

## Conclusions

Delayed or absent birth records could limit the government's ability to understand the population's needs and are associated with long-term health care use and outcomes. Lack of death registration and evidence of disparities in registration by sex could lead decision-makers target services inappropriately and lead demographers to make inaccurate estimations of trends. While support for those collecting this information is crucial, addressing socio-cultural beliefs and incentives for the death registration of certain people (infants, women) is also critical. Adequate awareness campaign is required to increase birth and death registration level in Bihar.

## Supporting information

**S1 Checklist. COREQ checklist.**
(DOC)

**S1 Text. Guide for KII and FGD.**
(DOCX)

**S1 Table. Rate (%) of health and education indicators in the Bihar and India, 2015–2016.**
(DOCX)

**S2 Table. Names of blocks of Patna and Vaishali.**
(XLSX)

**S3 Table. Estimates of birth and death registration level by districts of Bihar.**
(XLS)

## Acknowledgments

The authors would like to acknowledge study participants who co-operated and provided their valuable time for the in-depth interview and FGD. Besides, we are thankful to all the Ambedkar Library staff, Jawaharlal Nehru University, New Delhi, for providing access to Journals.

## Author Contributions

**Conceptualization:** Krishna Kumar, Nandita Saikia.

**Data curation:** Krishna Kumar.

**Formal analysis:** Krishna Kumar, Nandita Saikia.

**Investigation:** Krishna Kumar, Nandita Saikia.

**Methodology:** Krishna Kumar, Nandita Saikia, Nadia Diamond-smith.

**Supervision:** Nandita Saikia, Nadia Diamond-smith.

**Validation:** Nandita Saikia.

**Visualization:** Nadia Diamond-smith.

**Writing – original draft:** Krishna Kumar.

**Writing – review & editing:** Nandita Saikia, Nadia Diamond-smith.

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
