## [Decision Letter · Decision Letter 0]

21 Jan 2022

PONE-D-21-33029Performance barriers of Civil Registration System in Bihar: An exploratory studyPLOS ONE

Dear Dr. Saikia,

Thank you for submitting your manuscript to PLOS ONE. After careful consideration, we feel that it has merit but does not fully meet PLOS ONE’s publication criteria as it currently stands. Therefore, we invite you to submit a revised version of the manuscript that addresses the points raised during the review process. Please submit your revised manuscript by Mar 07 2022 11:59PM. If you will need more time than this to complete your revisions, please reply to this message or contact the journal office at plosone@plos.org. Please include the following items when submitting your revised manuscript:A rebuttal letter that responds to each point raised by the academic editor and reviewer(s). You should upload this letter as a separate file labeled 'Response to Reviewers'.A marked-up copy of your manuscript that highlights changes made to the original version. You should upload this as a separate file labeled 'Revised Manuscript with Track Changes'.An unmarked version of your revised paper without tracked changes. You should upload this as a separate file labeled 'Manuscript'.

We look forward to receiving your revised manuscript.

Kind regards,

Bidhubhusan Mahapatra, Ph.D.

Academic Editor

PLOS ONE

Journal Requirements:

4. We note that Figure 2 in your submission contain [map/satellite] images which may be copyrighted. All PLOS content is published under the Creative Commons Attribution License (CC BY 4.0), which means that the manuscript, images, and Supporting Information files will be freely available online, and any third party is permitted to access, download, copy, distribute, and use these materials in any way, even commercially, with proper attribution. For these reasons, we cannot publish previously copyrighted maps or satellite images created using proprietary data, such as Google software (Google Maps, Street View, and Earth). For more information, see our copyright guidelines: http://journals.plos.org/plosone/s/licenses-and-copyright.

Additional Editor Comments:

This is an important area of work; however, there are areas that can be strengthened further. I suggest the authors to take a careful look at the very useful suggestions from the reviewers and revise the paper accordingly.

Reviewers' comments:

Reviewer's Responses to Questions

**Comments to the Author**

1. Is the manuscript technically sound, and do the data support the conclusions?

Reviewer #1: Yes

Reviewer #2: Partly

Reviewer #3: Partly

2. Has the statistical analysis been performed appropriately and rigorously? 

Reviewer #1: Yes

Reviewer #2: N/A

Reviewer #3: N/A

3. Have the authors made all data underlying the findings in their manuscript fully available?

Reviewer #1: Yes

Reviewer #2: Yes

Reviewer #3: Yes

4. Is the manuscript presented in an intelligible fashion and written in standard English?

Reviewer #1: Yes

Reviewer #2: Yes

Reviewer #3: Yes

5. Review Comments to the Author

Reviewer #1: This is a good piece of scientific work addressing a very important yet neglected issue. Good efforts have been put together to develop this manuscript though it needs some major revisions.

Major points:

1. Line Nos 121-129: Please make study rationale more clear

2. Figure 1:

a. How are the deaths at 'home' in 'urban' areas registered? Missing in the diagram.

b. How are the deaths in 'hospitals' in rural areas registered? missing in the diagram.

c. Is ASHA/ANM playing any role in death registration as being currently used in Punjab and Haryana? if yes, then mention please. Who are the designated key informants to whom family members can report FIR of an event in both urban and rural areas?

d. At district and state level, which department is responsible for births and deaths registration? Is it Health department like in Haryana and Punjab? These points are not coming out explicitly from the diagram itself!!

3. Please mention three things in each box of Figure 3: Designation of the person under CRS, name of the department the person holds and his/her designation in that department. Example: Like you have mentioned Chief Registrar is the Director of Statistics and Evaluation but of which department???? Please elaborate

4. Did you consider the levels of birth and death registration of district and block as selection criteria for sampling units because that might give a better picture of reasons.

5. The methodology is indicating that the study was conducted among rural aspect of CRS as none of the study participants represented the urban population or municipal corporation. This has undermined the study findings to rural area only. I feel you need to explain this as a study limitation and provide a valid reason for not including the urban representation.

6. In the methodology section, it's important to mention about the levels of birth and death registrations of two districts you have selected. and compare it with other districts.

Minor revisions:

1. Please use one terminology throughout either CRS or CRVS in introduction section

2. Expand ORGI in caption of Figure 1

3. Please correct the spelling of 'anganwadi sevika' throughout the manuscript

4. Mention inthe data collection part of methodology section about the total number of FGDs done

5. Mention inthe data collection part of methodology section about the total number of KII done

6. Line nos 290-93 not clear

7. Please provide supporting data for statement no. 528-529

8. Please provide supporting data for statement no. 338-39

9. Please add references for lines 182-186

10. In line no. 556-57, you have mentioned that birth and death registration is not being done at sub-center or village levle? Is it not community based as of now? Revoew the lines

11. You have mentioned in line 567 that all PHCs are not linked to CRS. Is this statement true? Please review.

Reviewer #2: Review: Performance barriers of Civil Registration System in Bihar: An exploratory study

This is an interesting exploratory study, relying on interviews with government officials and rural residents, which documents the functioning of the civil registration system in Bihar before the pandemic. Given the importance of understanding and improving the civil registration system, such studies are urgently needed. I felt that the study should have been more in-depth. Interviews with CRS officials and with community members could have been more detailed, and the description of the overall state of the CRS in Bihar, despite the authors’ attempts, is still quite incomplete. However, given the paucity of scholarship on the civil registration system in India, specially of a qualitative nature, this paper is welcome. I have some minor suggestions, which I hope would be helpful for the authors in revising their work.

1. The authors motivate their paper saying that it is important to study the civil registration system in Bihar because of its poor development indicators in general, and the fact that its CRS is estimated to record the lowest proportion of deaths and births. However, although the authors provide some hints at why Bihar’s CRS performs so poorly, we do not get an explicit discussion of why precisely Bihar lacks even other northern Indian states, such as Madhya Pradesh or Rajasthan, which have higher registration coverage than Bihar. In this, the state of Bihar and Uttar Pradesh are way behind all the other states in India, and it is not clear why this is the case. To truly understand this, a comparative design is needed. Perhaps the case is that the lack of linkages between health systems and the CRS which the authors document, or the lack of computerization in Bihar explains why it lags behind other Indian states (even similarly poor ones). My suggestion to the authors is to consider this line of thinking carefully. They can see what the extant literature says about other poor Indian states (they may find looking at the annual CRS reports at the state level and the national level useful here) about CRS being so poor in Bihar. They should also note some lines of thinking for future researchers in the discussion section, as well as perhaps a limitation that they have not adequately addressed these questions.

2. Related to this point, we also don’t get a sense of the changes in the Civil Registration System before the pandemic in Bihar. Existing patterns, both from the data that has been accessed by journalists during the pandemic period (for instance see https://www.indiacovidmapping.org/reports/mortality/BiharFactsheet.pdf), as well as annual estimates before the pandemic suggest improvements in the Civil Registration System in Bihar. How did these improvements came about? If the authors have information on this, it would be extremely valuable. Similarly, what happened to Bihar’s civil registration system during the pandemic? Answering this likely requires fresh data collection. If that is not feasible, then that should be noted in the discussion. If it is possible to call CRS staff on phones and ask some of these questions, then that would be quite valuable. If this is not possible, then perhaps the authors can consider this as a future research endeavor.

3. At a large number of places, I felt that the authors have policy recommendations that were not supported by the information they present. In some cases, their recommendations can actually be harmful. For instance, the authors recommend “mandatory burial or cremation permits.” Instead of improving registration, I can see how this kind of procedure would cause even more problems in a place like Bihar. For instance, people will have to wait to get a death registered and could then only approach burial or cremation grounds? Instead of this, what would work well is a “notification” and “proof” system: burial grounds can notify CRS authorities about a death, and the family could have a slip which would help as a proof of death. These linkages exist in other states, especially with healthcare facilities. Public hospitals can in fact issue death certificates in many states, and private hospitals issue a hospital death certificate (or birth certificate) which can be taken to a registrar to get a death registered.

I was also uncomfortable with the recommendation of fines (lines 533-534). First, the authors already say that there is a fine of Rs. 10 to register a death after 21 days. Worse, the procedure described here to register a death after 21 days looks like a nightmare, with visits to far off offices. More fines when clearly it is the government system that is lacking is a bit counterproductive. The authors should remove this recommendation. In any case, this does not follow from the data presented by the authors.

4. I found the sentence “there was no strict implementation of the birth and death registration law” vague. Which aspects of the law were not strictly implemented, and how was this ascertained?

5. I was not convinced by the literature cited that “lack of birth registration is associated with school dropout …” and so on. It is quite likely that this is just selection, for instance poor children are more likely to drop out and poor children are less likely to have had their births registered. I would suggest dropping this citation.

6. The authors cite the CRS report to report that 7 percent of births and 8 percent of deaths were not registered in 2019. Although this is an official estimate, it is likely an over-estimate. There are multiple reasons for this, among them the fact that the 2019 CRS deaths in this report are compared to 2018 SRS. Other reasons include the fact that the SRS may under-estimate the crude death rate. This line of thinking is explored in detail here (appendix 1: https://www.medrxiv.org/content/10.1101/2021.09.30.21264376v1). I leave it to the authors to decide how best to convey the uncertainty in the true rate of registration completion. Perhaps the authors can say “according to official estimates …”.

7. In figure 1, which provides the organization of the CRS, I was surprised that the Panchayat was not mentioned anywhere in the hierarchy. In figure 3, the panchayat sevak is mentioned as a registrar. In some other northern states, such as Madhya Pradesh, the bulk of rural death and birth registration happens at the Panchayat level. Is it the case that this does not happen in Bihar? In Madhya Pradesh, I have seen functioning computers in Panchayat offices which can register deaths. In which case, this may be the reason why death registration is so poor in Bihar – the lack of panchayat registration. The authors could perhaps explore this more, and correct figure 1 if necessary. If the panchayat is not a functional place to register deaths, then this is actually an important policy suggestion, to improve registration and computerization at the Panchayat level.

8. Figure 1 claims that an annual report at the “state level is published by the chief registrar” and also that “Annual report at India and state level is published by ORGI”. This is confusing, and likely the latter sentence needs modification.

9. I thought that the bit about people not reporting infant deaths because it is considered a sin of parents in previous birth was quite revealing. The state government should consider potentially challenging this belief, and this also is a policy recommendation that flows from the findings from the paper. In addition, given the levels of awareness that the authors document, perhaps more information campaigns are also worth recommending.

10. Similarly, I was surprised by the sentence that “Public facilities record all birth and death but don't update on the CTS portal until birth and death certificates are requested by family members.” The authors do not present evidence for it. This is quite important, and also deserves to have highlighting from the perspective of policy recommendation that the paper generates.

11. In line 536: the authors say the CRS form is complex. Perhaps they should elaborate more? I do not agree with all the things the CRS form asks (for instance, whether the person smoked or drank alcohol is not collected by most other countries) but I didn’t have the impression that it was very complex. If the authors think it is complex, I would like to see an explanation why. Otherwise, I would like this sentence to be reformulated.

Reviewer #3: I am reviewing this paper from a non-expert perspective - I have limited knowledge of the civil registration system itself, and the specific issues faced in Bihar.

The paper clearly explores an important issue, namely barriers to civil registration in Bihar. The importance of strengthening civil registration has been highlighted by the pandemic, making this study timely. The methodology - interviews with key informants and members of the public - seems appropriate, and the paper includes a number of interesting insights about these barriers at both administrative and cultural levels.

I have a few comments for the authors to consider:

1. Quantitative data. Although this is a qualitative study, it would still be helpful to have some summary quantitative data. At least: amongst the focus group participants in Patna how many births and how many deaths were reported; and how many of these were registered. Plus the same for Vaishali. This is not to draw any quantitative conclusions, but merely to understand the overall sample.

In fact, some quantitative data is presented, but too unclearly. "A few participants (n=15) reported death at the registration centres situated at cemeteries or burial places" - this sentence needs a denominator: 15 out of how many? In sentences such as "Besides, two-fifth of the study participants reported the death of their household members at the block or municipality offices." it would be better to present the actual numbers of how many deaths were reported, and out of these how many at block or municipality offices.

The same comments apply at several other places in the text, for example: "Nearly one-sixth of the total participants were unaware of the death registration procedures and benefits associated with death registration. A higher number of participants from Patna applied for birth and death registration of their household members than participants from Vaishali. Rural and urban differentials in birth and death registration were found." In every case numbers, with denominators, would be better.

"However, a subset of participants (n=10) did not receive a birth certificate for their child at all." Again, a denominator is needed. Also, not clear: were these people who applied for, but never received, the birth certicificate; or who never applied for the certificate?

2. Context and history of death registration in the state. The authors mention inconsistency in death registration in Bihar - the lack of a clear trend (up to 2019 at least). Do they have any insights as to why there is so much year-to-year variation in the estimated levels of death registration? Are the SRS-based estimates of coverage reliable in their view? What are the factors which could cause a significant drop in registration from one year to the next? If there were changes in the systems in place which affected trends in registration, then this would be interesting to know. Did the interviews provide any hints?

3. Poverty and marginalisation. I find these sentences problematic: "Lack of birth registration is associated with increased school dropout, child trafficking, and child labor [1,12]. The child mortality rate is higher among unregistered children [13]." It is very likely that the common factor behind these associations is poverty and marginalisation - the sentences are not formally wrong, but suggest the lack of birth registration could be a causal factor behind, say, child trafficking or infant mortality. It seems important to be more clear here.

This raises an important point: the authors really should discuss how factors such as caste and poverty are associated with levels of registration. Are there studies on this topic, relevant to Bihar? If there is a relevant literature, then it should be referenced and discussed at least briefly. It would be interesting to know whether there are examples of other states where poverty and marginalisation are comparable to Bihar, but civil registration is higher and more stable, and if the authors have some insight about why. If many people in Bihar perceive little benefit in reporting deaths, then why is this different in other comparable states? Are the practical barriers - loss of time and income, difficulty, etc., different in other states?

Although the sample may be too small to understand how attitudes to registration vary by caste/class there still seem to be some hints in the data (e.g. not registering infant deaths - "sin of previous birth"). It seems that the authors did not collect demographic data on caste/ occupation/ socioeconomic status, etc? Perhaps the authors should mention this as a limitation of the study. It would still be interesting if they have any insights from this work or the work of others on how these factors might affect access to registration or attitudes to registration.

4. Recommendations. A number of the recommendations for how to improve vital registration are made by the authors, and many of these could be useful. But it is not really clear where responsibility lies for the failings. After statements like "the officers did not take any steps for improving the registration level in recent times, particularly death registration." could the authors indicate what the officers could have done? Also perhaps there could be more discussion of resources. A "lack of adequate investment" is mentioned in the abstract, and later the key informants "...suggested that adequate funds should be provided for the smooth functioning of the system." What is the resourcing of the CRS in Bihar? Did the interviews provide insight into where such funds should go (more staff? better pay? better training? more registration offices?) Here, again, some comparison with other states would be interesting.

5. Minor points

"Epidemiological purposes" are mentioned in the introduction - the authors could say more about how, in the context of the current pandemic, there has been an urgent need for all cause mortality data on account of weak official surveillance of COVID mortality. Any comments on uncertainties around pandemic mortality in Bihar and in comparable states would help to highlight the current importance of this study.

In Table S1, the fertility rates for Bihar and India appear to be exchanged.

Table S1 should perhaps include the estimated death registration level based on SRS and/or NFHS?

I would suggest the authors carefully reread the manuscript for typographical errors.

6. PLOS authors have the option to publish the peer review history of their article (what does this mean?). If published, this will include your full peer review and any attached files.

Reviewer #1: **Yes: **Mamta Gupta

Reviewer #2: **Yes: **Aashish Gupta

Reviewer #3: No

---

## [Author Response · Author response to Decision Letter 0]

16 Mar 2022

Reply to respected reviewers.

Reviewer #1: This is a good piece of scientific work addressing a very important yet neglected issue. Good efforts have been put together to develop this manuscript though it needs some major revisions.

Major points:

1. Line Nos 121-129: Please make the study rationale more clear

Reply- We have revised the manuscript in light of your suggestion. Please see line no. 121-130.

2. Figure 1:

a. How are the deaths at 'home' in 'urban' areas registered? Missing in the diagram.

Reply- We have revised the diagram in light of your suggestion. Please see page no.5, fig. 1 

b. How are the deaths in 'hospitals' in rural areas registered? missing in the diagram.

Reply- We have revised the diagram in light of your suggestions. Deaths in Hospitals are registered by the head of the institution (Chief Medical Officer). Please see page no.5, fig 1. 

c. Is ASHA/ANM playing any role in death registration as being currently used in Punjab and Haryana? if yes, then mention, please. Who are the designated key informants to whom family members can report FIR of an event in both urban and rural areas?

Reply- In Bihar, ASHA/ANM does not play any role in birth and death registration.

We have revised the manuscript now. Please see page no. 5, fig 1

d. At the district and state level, which department is responsible for births and deaths registration? Is it the Health department like in Haryana and Punjab? These points are not coming out explicitly from the diagram itself!!

Reply- Registrar at Nagar Nigam, District registrar office, and Chief medical officer of district-level hospitals are mainly responsible for birth and death registration. At the state level, Chief registrar, Joint Chief Registrar, and Dy. Chief Registrar (Vital Statistics) is responsible for birth and death registration. We have revised fig 3. (Page no. 7)

3. Please mention three things in each box of Figure 3: Designation of the person under CRS, name of the department the person holds, and his/her designation in that department. Example: Like you have mentioned Chief Registrar is the Director of Statistics and Evaluation but of which department???? Please elaborate

Reply- Chief Registrar is the Director of the Directorate of Statistics and Evaluation belonging to the Planning and Development Department in Bihar. We have revised fig 3 in light of your suggestion. Please see fig3, Page no. 7

4. Did you consider the levels of birth and death registration of district and block as selection criteria for sampling units because that might give a better picture of reasons.

Reply- We observed district-level death registration level as selection criteria in addition to urbanization rate. We purposively selected a district with higher death registration and a district with lower death registration. Furthermore, four blocks are chosen randomly from each district. We have revised the write-up now. Please see line no. 153-159.

5. The methodology is indicating that the study was conducted among the rural aspect of CRS as none of the study participants represented the urban population or municipal corporation. This has undermined the study findings to rural areas only. I feel you need to explain this as a study limitation and provide a valid reason for not including the urban representation.

Reply- We also included considerable study participants from urban areas. We have presented the methodology more clearly now. Please see line no. 170-177.

6. In the methodology section, it's important to mention the levels of birth and death registrations of the two districts you have selected. and compare it with other districts.

Reply- We have included the death registration level of two selected districts (Patna and Vaishali) in methodology (lines no. 157 & 159). We have provided a supplementary table showing birth and death registration levels by the districts of Bihar. Please see supplementary file S3 Table.

Minor revisions:

1. Please use one terminology throughout either CRS or CRVS in the introduction section

Reply- We have revised the manuscript in light of your valuable suggestions. We are using CRS throughout the manuscript. 

2. Expand ORGI in the caption of Figure 1

Reply- We have revised the manuscript. We have written a different title now. ” Fig-1 Flow chart of birth and death registration in Bihar” (Please see page no. 5)

3. Please correct the spelling of 'Anganwadi sevika' throughout the manuscript

Reply- We have corrected the spelling of Anganwadi sevika in the revised manuscript.

4. Mention in the data collection part of the methodology section about the total number of FGDs done

Reply- We have included the total number of FGD done in the revised manuscript. Please see line no.- 183

5. Mention in the data collection part of the methodology section about the total number of KII done

Reply- We have included the total number of KII done in the revised manuscript. Please see line no. 183 

6. Line nos. 290-93 not clear

Reply- We have revised the manuscript now. Please see line no. 248-253

7. Please provide supporting data for statement no. 528-529

Reply- We have included the supporting data now. Please see line no. 505

8. Please provide supporting data for statement no. 338-39

Reply- We have found lower reporting for death, particularly, infant and female death, in Bihar. We provided the transcript along with this manuscript. A study also showed death registration coverage is poor in Bihar (Reference no. 38 in the manuscript)

9. Please add references for lines 182-186

Reply- We have included a supplementary table (S3 table) in support of lines 182-186.

10. In line no. 556-57, you have mentioned that birth and death registration is not being done at sub-center or village level? Is it not community-based as of now? Review the lines

Reply- In rural areas, birth and death occur at home are registered by Anganwadi sevika within 21 days of the event. In urban areas, birth and death occur at home is registered by registrar at municipal corporation and block offices. Institutional birth and death is registered by designated health staff at health centers within 21 days of events. Please see fig 1, page no. 5. We have revised the manuscript now. 

11. You have mentioned in line 567 that all PHCs are not linked to CRS. Is this statement true? Please review.

Reply- We have found many PHC are not linked to CRS due to lack of adequate logistics and lack of dedicated staff or negligence by the medical officer. A recent paper also showed registration are missed in PHC due to negligence (Kumar et al., 2021) 

Reviewer #2: Review: Performance barriers of Civil Registration System in Bihar: An exploratory study

This is an interesting exploratory study, relying on interviews with government officials and rural residents, which documents the functioning of the civil registration system in Bihar before the pandemic. Given the importance of understanding and improving the civil registration system, such studies are urgently needed. I felt that the study should have been more in-depth. Interviews with CRS officials and with community members could have been more detailed, and the description of the overall state of the CRS in Bihar, despite the authors’ attempts, is still quite incomplete. However, given the paucity of scholarship on the civil registration system in India, specially of a qualitative nature, this paper is welcome. I have some minor suggestions, which I hope would be helpful for the authors in revising their work.

1. The authors motivate their paper saying that it is important to study the civil registration system in Bihar because of its poor development indicators in general, and the fact that its CRS is estimated to record the lowest proportion of deaths and births. However, although the authors provide some hints at why Bihar’s CRS performs so poorly, we do not get an explicit discussion of why precisely Bihar lacks even other northern Indian states, such as Madhya Pradesh or Rajasthan, which have higher registration coverage than Bihar. In this, the state of Bihar and Uttar Pradesh are way behind all the other states in India, and it is not clear why this is the case. To truly understand this, a comparative design is needed. Perhaps the case is that the lack of linkages between health systems and the CRS which the authors document, or the lack of computerization in Bihar explains why it lags behind other Indian states (even similarly poor ones). My suggestion to the authors is to consider this line of thinking carefully. They can see what the extant literature says about other poor Indian states (they may find looking at the annual CRS reports at the state level and the national level useful here) about CRS being so poor in Bihar. They should also note some lines of thinking for future researchers in the discussion section, as well as perhaps a limitation that they have not adequately addressed these questions.

Reply- There are limited studies on CRS at state levels, particularly in Bihar. We could not locate any study examining why CRS is poor in Bihar. Although we welcome the reviewer’s comment on a comparative picture of Bihar/UP with MP or Rajasthan, it is beyond the scope of the current study. Therefore, a comparative study cannot be done for the present study as it collected information of Bihar only. We may explore this research gap in future studies. We have included these lines in the limitation section of this manuscript. Please see line no. 560-561.

2. Related to this point, we also don’t get a sense of the changes in the Civil Registration System before the pandemic in Bihar. Existing patterns, both from the data that has been accessed by journalists during the pandemic period (for instance see https://www.indiacovidmapping.org/reports/mortality/BiharFactsheet.pdf), as well as annual estimates before the pandemic suggest improvements in the Civil Registration System in Bihar. How did these improvements came about? If the authors have information on this, it would be extremely valuable. Similarly, what happened to Bihar’s civil registration system during the pandemic? Answering this likely requires fresh data collection. If that is not feasible, then that should be noted in the discussion. If it is possible to call CRS staff on phones and ask some of these questions, then that would be quite valuable. If this is not possible, then perhaps the authors can consider this as a future research endeavor.

Reply- We have included information regarding development of CRS in Bihar before pandemic. Also, we added what happen to CRS in Bihar during the pandemic in the revised manuscript. Please see line no. 452-455.& 459-466. 

3. At a large number of places, I felt that the authors have policy recommendations that were not supported by the information they present. In some cases, their recommendations can actually be harmful. For instance, the authors recommend “mandatory burial or cremation permits.” Instead of improving registration, I can see how this kind of procedure would cause even more problems in a place like Bihar. For instance, people will have to wait to get a death registered and could then only approach burial or cremation grounds? Instead of this, what would work well is a “notification” and “proof” system: burial grounds can notify CRS authorities about a death, and the family could have a slip which would help as a proof of death. These linkages exist in other states, especially with healthcare facilities. Public hospitals can in fact issue death certificates in many states, and private hospitals issue a hospital death certificate (or birth certificate) which can be taken to a registrar to get a death registered.

I was also uncomfortable with the recommendation of fines (lines 533-534). First, the authors already say that there is a fine of Rs. 10 to register a death after 21 days. Worse, the procedure described here to register a death after 21 days looks like a nightmare, with visits to far off offices. More fines when clearly it is the government system that is lacking is a bit counterproductive. The authors should remove this recommendation. In any case, this does not follow from the data presented by the authors.

Reply- We have revised the manuscript in light of your valuable suggestions. We have deleted mandatory registration burial or cremation and fine for late registration.

4. I found the sentence “there was no strict implementation of the birth and death registration law” vague. Which aspects of the law were not strictly implemented, and how was this ascertained?

Reply- We found health institution does delayed registration; however, it is not allowed as per official guidelines. These event should be registered by Block Development Officer or District Statistical Officer in such cases. A similar problems were also shown by a previous literature ( Kumar et al; 2021).We have included this sentence in revised manuscript. Please see line no. 433-434.

5. I was not convinced by the literature cited that “lack of birth registration is associated with school dropout …” and so on. It is quite likely that this is just selection, for instance poor children are more likely to drop out and poor children are less likely to have had their births registered. I would suggest dropping this citation.

Reply-We tried to show the association of unregistered birth and school dropout. We don’t intend to show absence of birth certificate is a causal factor of the school drop-out. However, we have reframed the sentence now. Please see line no. 91-92

6. The authors cite the CRS report to report that 7 percent of births and 8 percent of deaths were not registered in 2019. Although this is an official estimate, it is likely an over-estimate. There are multiple reasons for this, among them the fact that the 2019 CRS deaths in this report are compared to 2018 SRS. Other reasons include the fact that the SRS may under-estimate the crude death rate. This line of thinking is explored in detail here (appendix 1: https://www.medrxiv.org/content/10.1101/2021.09.30.21264376v1). I leave it to the authors to decide how best to convey the uncertainty in the true rate of registration completion. Perhaps the authors can say “according to official estimates …”.

Reply- We have written “According to official statistics”, line no. 95-96. 

7. In figure 1, which provides the organization of the CRS, I was surprised that the Panchayat was not mentioned anywhere in the hierarchy. In figure 3, the panchayat sevak is mentioned as a registrar. In some other northern states, such as Madhya Pradesh, the bulk of rural death and birth registration happens at the Panchayat level. Is it the case that this does not happen in Bihar? In Madhya Pradesh, I have seen functioning computers in Panchayat offices which can register deaths. In which case, this may be the reason why death registration is so poor in Bihar – the lack of panchayat registration. The authors could perhaps explore this more, and correct figure 1 if necessary. If the panchayat is not a functional place to register deaths, then this is actually an important policy suggestion, to improve registration and computerization at the Panchayat level.

Reply- Panchayat secretary office registers birth and death after 21 days of the occurrence of event in rural areas in Bihar. I have revised the fig 1. Please see page no. 5

8. Figure 1 claims that an annual report at the “state level is published by the chief registrar” and also that “Annual report at India and state level is published by ORGI”. This is confusing, and likely the latter sentence needs modification.

Reply- Annual report at India level is published by ORGI, New Delhi. In Bihar, state level report is published by Directorate of Economic and Statistics, Department of planning and development. We have changed the fig 1 now. (Please see page no .5)

9. I thought that the bit about people not reporting infant deaths because it is considered a sin of parents in previous birth was quite revealing. The state government should consider potentially challenging this belief, and this also is a policy recommendation that flows from the findings from the paper. In addition, given the levels of awareness that the authors document, perhaps more information campaigns are also worth recommending.

Reply- We have revised the manuscript in light of suggestions. Please see line no. 544-45

10. Similarly, I was surprised by the sentence that “Public facilities record all birth and death but don't update on the CTS portal until birth and death certificates are requested by family members.” The authors do not present evidence for it. This is quite important, and also deserves to have highlighting from the perspective of policy recommendation that the paper generates.

Reply- We observed health staff don't update registered data from registers to the CRS portal until birth and death certificates are requested by family members. A study documented a similar problem (Reference no. 38). We recommended for proper monitoring of all registration units. Please see line no. 548

11. In line 536: the authors say the CRS form is complex. Perhaps they should elaborate more? I do not agree with all the things the CRS form asks (for instance, whether the person smoked or drank alcohol is not collected by most other countries) but I didn’t have the impression that it was very complex. If the authors think it is complex, I would like to see an explanation why. Otherwise, I would like this sentence to be reformulated.

Reply- We have removed this sentence in the revised manuscript. 

Reviewer #3: I am reviewing this paper from a non-expert perspective - I have limited knowledge of the civil registration system itself, and the specific issues faced in Bihar.

The paper clearly explores an important issue, namely barriers to civil registration in Bihar. The importance of strengthening civil registration has been highlighted by the pandemic, making this study timely. The methodology - interviews with key informants and members of the public - seems appropriate, and the paper includes a number of interesting insights about these barriers at both administrative and cultural levels.

I have a few comments for the authors to consider:

1. Quantitative data. Although this is a qualitative study, it would still be helpful to have some summary quantitative data. At least: amongst the focus group participants in Patna how many births and how many deaths were reported; and how many of these were registered. Plus the same for Vaishali. This is not to draw any quantitative conclusions, but merely to understand the overall sample.

In fact, some quantitative data is presented, but too unclearly. "A few participants (n=15) reported death at the registration centres situated at cemeteries or burial places" - this sentence needs a denominator: 15 out of how many? In sentences such as "Besides, two-fifth of the study participants reported the death of their household members at the block or municipality offices." it would be better to present the actual numbers of how many deaths were reported, and out of these how many at block or municipality offices.

The same comments apply at several other places in the text, for example: "Nearly one-sixth of the total participants were unaware of the death registration procedures and benefits associated with death registration. A higher number of participants from Patna applied for birth and death registration of their household members than participants from Vaishali. Rural and urban differentials in birth and death registration were found." In every case numbers, with denominators, would be better.

"However, a subset of participants (n=10) did not receive a birth certificate for their child at all." Again, a denominator is needed. Also, not clear: were these people who applied for, but never received, the birth certicificate; or who never applied for the certificate?

Reply- Dear reviewer, we have not explored categorically the number of registered births and deaths, in this study. However, we have shown how many participants registered birth and death of their family members. Besides, we have included denominator in the revised manuscript in light of your suggestions.

2. Context and history of death registration in the state. The authors mention inconsistency in death registration in Bihar - the lack of a clear trend (up to 2019 at least). Do they have any insights as to why there is so much year-to-year variation in the estimated levels of death registration? Are the SRS-based estimates of coverage reliable in their view? What are the factors which could cause a significant drop in registration from one year to the next? If there were changes in the systems in place which affected trends in registration, then this would be interesting to know. Did the interviews provide any hints?

Reply. Our primary study revealed that birth and death registration was decreased during 2013 and 2014 due to majority of CRS staff are engaged in Bihar Assembly election duty. Thereafter, there has been continuous increase in birth and death registration due to introduction of digital registration using CRS portal and awareness campaign.

3. Poverty and marginalisation. I find these sentences problematic: "Lack of birth registration is associated with increased school dropout, child trafficking, and child labor [1,12]. The child mortality rate is higher among unregistered children [13]." It is very likely that the common factor behind these associations is poverty and marginalisation - the sentences are not formally wrong, but suggest the lack of birth registration could be a causal factor behind, say, child trafficking or infant mortality. It seems important to be more clear here.

This raises an important point: the authors really should discuss how factors such as caste and poverty are associated with levels of registration. Are there studies on this topic, relevant to Bihar? If there is a relevant literature, then it should be referenced and discussed at least briefly. It would be interesting to know whether there are examples of other states where poverty and marginalisation are comparable to Bihar, but civil registration is higher and more stable, and if the authors have some insight about why. If many people in Bihar perceive little benefit in reporting deaths, then why is this different in other comparable states? Are the practical barriers - loss of time and income, difficulty, etc., different in other states?

Although the sample may be too small to understand how attitudes to registration vary by caste/class there still seem to be some hints in the data (e.g. not registering infant deaths - "sin of previous birth"). It seems that the authors did not collect demographic data on caste/ occupation/ socioeconomic status, etc? Perhaps the authors should mention this as a limitation of the study. It would still be interesting if they have any insights from this work or the work of others on how these factors might affect access to registration or attitudes to registration.

Reply- We have shown association in the sentence “Lack of birth registration is associated with increased school dropout, child trafficking, and child labor”, not the causal factor. Moreover, we have reframed this sentence in the revised manuscript “A low birth registration is associated with increased school dropout, child trafficking, and child labor”. We did not collect socio-economic data in this study. Also, comparison study with other similar state cannot be done at present due to limited literature. We have mention this sentence in limitation section of this study. Please see line no. -559-560

4. Recommendations. A number of the recommendations for how to improve vital registration are made by the authors, and many of these could be useful. But it is not really clear where responsibility lies for the failings. After statements like "the officers did not take any steps for improving the registration level in recent times, particularly death registration." could the authors indicate what the officers could have done? Also perhaps there could be more discussion of resources. A "lack of adequate investment" is mentioned in the abstract, and later the key informants "...suggested that adequate funds should be provided for the smooth functioning of the system." What is the resourcing of the CRS in Bihar? Did the interviews provide insight into where such funds should go (more staff? better pay? better training? more registration offices?) Here, again, some comparison with other states would be interesting.

Reply- We have revised the manuscript in light of the suggestions. However, there is limited study on CRS in India, so a comparative study cannot be done at this stage. We will explore this issue in a future study. We have included some suggestions on the role of authority in the recommendation section of this manuscript. Please see line no. 543-551

5. Minor points

"Epidemiological purposes" are mentioned in the introduction - the authors could say more about how, in the context of the current pandemic, there has been an urgent neall-causel cause mortality data on account of weak official surveillance of COVID mortality. Any comments on uncertainties around pandemic mortality in Bihar and in comparable states would help to highlight the current importance of this study.

Reply- We have revised the manuscript in view of suggestions. Please see line no. 452-455

In Table S1, the fertility rates for Bihar and India appear to be exchanged.

Reply- We have cored the Table S1 .Please see supplementary table (S1 table).

Table S1 should perhaps include the estimated death registration level based on SRS and/or NFHS?

Reply- We have included the official estimate of death registration level (Reference-Office of Registrar General of India (ORGI), Report, 2018). ORGI also use SRS death rate for calculating death registration level.

I would suggest the authors carefully reread the manuscript for typographical errors.

Reply- We have revised the manuscript now.

---

## [Decision Letter · Decision Letter 1]

25 Apr 2022

PONE-D-21-33029R1Performance barriers of Civil Registration System in Bihar: An exploratory studyPLOS ONE

Dear Dr. Saikia,

Thank you for submitting your manuscript to PLOS ONE. After careful consideration, we feel that it has merit but does not fully meet PLOS ONE’s publication criteria as it currently stands. Therefore, we invite you to submit a revised version of the manuscript that addresses the points raised during the review process.

We look forward to receiving your revised manuscript.

Kind regards,

Bidhubhusan Mahapatra, Ph.D.

Academic Editor

PLOS ONE

Journal Requirements:

Reviewers' comments:

Reviewer's Responses to Questions

**Comments to the Author**

1. If the authors have adequately addressed your comments raised in a previous round of review and you feel that this manuscript is now acceptable for publication, you may indicate that here to bypass the “Comments to the Author” section, enter your conflict of interest statement in the “Confidential to Editor” section, and submit your "Accept" recommendation.

Reviewer #1: All comments have been addressed

Reviewer #2: (No Response)

Reviewer #3: (No Response)

2. Is the manuscript technically sound, and do the data support the conclusions?

Reviewer #1: Yes

Reviewer #2: Partly

Reviewer #3: Yes

3. Has the statistical analysis been performed appropriately and rigorously? 

Reviewer #1: N/A

Reviewer #2: Yes

Reviewer #3: N/A

4. Have the authors made all data underlying the findings in their manuscript fully available?

Reviewer #1: Yes

Reviewer #2: No

Reviewer #3: (No Response)

5. Is the manuscript presented in an intelligible fashion and written in standard English?

Reviewer #1: Yes

Reviewer #2: No

Reviewer #3: Yes

6. Review Comments to the Author

Reviewer #1: Line no. 170-72- Here you are mentioning that 05 persons of given designations were considered for KII. And you had selected four blocks from each district that means 08 blocks in total. so there should have been 40 KIIs instead of 15? Clarify please.

Please mention the department and ministry in Figure 3 for all registrars as highlighted in the attached file

Line 209: If the questionnaire was semi-structured, then it will have both open and closed ended questions. It cannot be both semi-structured and open ended simultaneously!!

Line 242-43: Do you think awareness and knowledge of registration procedures is a barrier to well-functioning CRS? Please reframe the sentence correctly.

Line 246-47: Please write the theme which is acting as a critical barrier from these paragraphs.

Line 250 : How many delivered in public hospital?

Line 254: Requiremnt of I-card should be mentioned as "PERCEIVED BARRIER"

Line 297: Did you conducted any analysis for this? Please mention the data

Line 386-96: I think this is a perceived administrative barrier. May think to shift in theme 2 paragraph

Line 459-61: Please mention the birth and death registration level of bihar from the latest CRS report

Reviewer #2: Thank you for paying attention to my comments. I still have minor comments, that I believe will strengthen the paper.

- In the objectives (line 33-34), the authors say that "Among all India's states, Bihar showed a lower recording of birth and death in 2019." I think this is not clear, because it can be interpreted as Bihar's death rate being actually lower than other states. Perhaps its better to write: "Bihar had one of the lowest levels of mortality registration among India's states".

- I don't get the point in the results of "death registration was not mandatory for burial or cremation permits." This is as it should be. Waiting for death registration before burial or cremation would be a nightmarish scenario. I really would like this sentence to be removed (I made similar remarks about this appearing elsewhere in the manuscript in the first revision).

- "a dedicated staff shortage" - you mean shortage of dedicated staff, I am guessing.

- Fig 1: I am surprised that the aanganawadi sevika can register deaths in rural Bihar. this could actually be a nice system, but its not clear from the paper about the extent to which this functions.

- Fig 1: similarly, its surprising that for births and deaths in health institutions in Bihar, the health system doesn't generate a birth certificate, and one still has to go to the panchayat secretary. In Fig 3, the authors mention the health system at local area level, but not in fig 1. again, this is quite confusing.

- It is revealing that the researchers visited the households with assistance from panchayat secretary. this could lead to biases in reporting, given the power a panchayat secretary has. I feel this should be mentioned as a limitation.

Reviewer #3: I think the paper is satisfactory and the revisions are adequate.

As a comment for the auhors, I still do feel that the authors could have attempted a summary of the data - e.g., a simple table of how many events (birth or death) occurred amongst the focus group participants, how many were registered (with/without delay), how many of the total births were registered through anganwadi sevikas/block officers, and so forth... this would have been helpful to set the scene. Sentences like "However, two-fifth of the total participants (n=90) applied for their child's birth registration after one year" remain ambiguous unless we know that births occurred in every family.

But perhaps it is not possible for the authors to be more precise if the data gathered does not allow this. And this comment is not critical to the qualitative conclusions of the manuscript.

7. PLOS authors have the option to publish the peer review history of their article (what does this mean?). If published, this will include your full peer review and any attached files.

Reviewer #1: **Yes: **Mamta Gupta

Reviewer #2: No

Reviewer #3: No

---

## [Author Response · Author response to Decision Letter 1]

27 Apr 2022

Reviewer #1: Line no. 170-72- Here you are mentioning that 05 persons of given designations were considered for KII. And you had selected four blocks from each district that means 08 blocks in total. so there should have been 40 KIIs instead of 15? Clarify please.

Reply- We thank you for your valuable observation. In this study, i wish to interview Block Statistical Officer (BSO), a Medical Officer, a Panchayat Chief or Ward Parsad (local leader), an Anganwadi Sevika and a registrar at Nagar Nigam as key informants from each block. However, many officials are not available during field study or did not consent for interview. Therefore, we interviewed total 15 key informants based on their availability and their consent for the interview for this study.

Please mention the department and ministry in Figure 3 for all registrars as highlighted in the attached file

Reply- We have revised the fig 3 in light of your suggestions. Please see page no. 7

Line 209: If the questionnaire was semi-structured, then it will have both open and closed ended questions. It cannot be both semi-structured and open ended simultaneously!!

Reply- We have removed the word “semi structured” in the revised manuscript. Please see line no. 209

Line 242-43: Do you think awareness and knowledge of registration procedures is a barrier to well-functioning CRS? Please reframe the sentence correctly.

Reply- We have reframed the sentence in the revised manuscript. Please see line no. 242-243

Line 246-47: Please write the theme which is acting as a critical barrier from these paragraphs.

Reply- We have revised the theme in the revised manuscript. Please see line no. 246-247

Line 250 : How many delivered in public hospital?

Reply- We have included statistics in the revised manuscript. Please see line no. 251.

Line 254: Requiremnt of I-card should be mentioned as "PERCEIVED BARRIER".

Reply- We have revised the manuscript in light of your valuable suggestion. Please see line no. 254-255

Line 297: Did you conducted any analysis for this? Please mention the data

Reply- We did not analyse death registration level by age and sex in this study. Therefore, we have removed this sentence in the revised manuscript. 

Line 386-96: I think this is a perceived administrative barrier. May think to shift in theme 2 paragraph

Reply- We have revised the manuscript in light of your valuable suggestion. Please see line no. 396-406.

Line 459-61: Please mention the birth and death registration level of bihar from the latest CRS report

Reply- We have revised the manuscript in light of your valuable suggestion. Please see line no. 464-465

Reviewer #2: Thank you for paying attention to my comments. I still have minor comments, that I believe will strengthen the paper.

- In the objectives (line 33-34), the authors say that "Among all India's states, Bihar showed a lower recording of birth and death in 2019." I think this is not clear, because it can be interpreted as Bihar's death rate being actually lower than other states. Perhaps its better to write: "Bihar had one of the lowest levels of mortality registration among India's states".

Reply- We have revised the manuscript in light of your valuable suggestion. Please see line no. 33-34

- I don't get the point in the results of "death registration was not mandatory for burial or cremation permits." This is as it should be. Waiting for death registration before burial or cremation would be a nightmarish scenario. I really would like this sentence to be removed (I made similar remarks about this appearing elsewhere in the manuscript in the first revision).

Reply- We have removed the sentence in light of your valuable suggestion. Please see line no. 45.

- "a dedicated staff shortage" - you mean shortage of dedicated staff, I am guessing.

Reply- Yes, we mean the same. However, we have reframed the words for clarity. Please see line no. 49

- Fig 1: I am surprised that the aanganawadi sevika can register deaths in rural Bihar. this could actually be a nice system, but its not clear from the paper about the extent to which this functions.

Reply- We have shown work profile of Aganwadi Worker in line no. 408-410. Besides, this study showed Aganwadi workers are one of the most active CRS staff in rural areas. 

- Fig 1: similarly, its surprising that for births and deaths in health institutions in Bihar, the health system doesn't generate a birth certificate, and one still has to go to the panchayat secretary. In Fig 3, the authors mention the health system at local area level, but not in fig 1. again, this is quite confusing.

Reply- We have revised fig 1 in the revised manuscript.

- It is revealing that the researchers visited the households with assistance from panchayat secretary. this could lead to biases in reporting, given the power a panchayat secretary has. I feel this should be mentioned as a limitation.

Reply- Panchayat Secretary only guided in finding the eligible households for this study. He had no contribution in FGD and participant’s consent for this study. We assume there was no considerable bias in the study due to panchayat secretary’s assistance.

Reviewer #3: I think the paper is satisfactory and the revisions are adequate.

As a comment for the auhors, I still do feel that the authors could have attempted a summary of the data - e.g., a simple table of how many events (birth or death) occurred amongst the focus group participants, how many were registered (with/without delay), how many of the total births were registered through anganwadi sevikas/block officers, and so forth... this would have been helpful to set the scene. Sentences like "However, two-fifth of the total participants (n=90) applied for their child's birth registration after one year" remain ambiguous unless we know that births occurred in every family.

But perhaps it is not possible for the authors to be more precise if the data gathered does not allow this. And this comment is not critical to the qualitative conclusions of the manuscript.

Reply- I thank you for your valuable suggestions. However, we did not collect detailed quantitative data. It is not possible to show the descriptive statistics of required variable in this study.

---

## [Decision Letter · Decision Letter 2]

10 May 2022

Performance barriers of Civil Registration System in Bihar: An exploratory study

PONE-D-21-33029R2

Dear Dr. Saikia,

We’re pleased to inform you that your manuscript has been judged scientifically suitable for publication and will be formally accepted for publication once it meets all outstanding technical requirements.

Kind regards,

Bidhubhusan Mahapatra, Ph.D.

Academic Editor

PLOS ONE

Additional Editor Comments (optional):

Reviewers' comments:

Reviewer's Responses to Questions

**Comments to the Author**

1. If the authors have adequately addressed your comments raised in a previous round of review and you feel that this manuscript is now acceptable for publication, you may indicate that here to bypass the “Comments to the Author” section, enter your conflict of interest statement in the “Confidential to Editor” section, and submit your "Accept" recommendation.

Reviewer #2: All comments have been addressed

2. Is the manuscript technically sound, and do the data support the conclusions?

Reviewer #2: Yes

3. Has the statistical analysis been performed appropriately and rigorously? 

Reviewer #2: N/A

4. Have the authors made all data underlying the findings in their manuscript fully available?

Reviewer #2: No

5. Is the manuscript presented in an intelligible fashion and written in standard English?

Reviewer #2: Yes

6. Review Comments to the Author

Reviewer #2: Thanks for addressing these comments. I look forward to seeing this paper in print. It would also be interesting to see follow up work on these topics. Thanks for your work!

7. PLOS authors have the option to publish the peer review history of their article (what does this mean?). If published, this will include your full peer review and any attached files.

Reviewer #2: No

---

## [Editor Report · Acceptance letter]

18 May 2022

PONE-D-21-33029R2 

Performance barriers of Civil Registration System in Bihar: An exploratory study 

Dear Dr. Saikia:

I'm pleased to inform you that your manuscript has been deemed suitable for publication in PLOS ONE. Congratulations! Your manuscript is now with our production department. 

Kind regards, 

on behalf of

Dr. Bidhubhusan Mahapatra 

Academic Editor

PLOS ONE